# Reconstruction of China's Farmland Rights System Based on the 'Trifurcation of Land Rights' Reform

**Linlin Li** [1,2,*], **Rong Tan** [1,2]  **and Cifang Wu** [1,2]

1   Department of Land Management, School of Public Affairs, Zhejiang University, Zijingang Campus, Hangzhou, 310058, China; tanrong@zju.edu.cn (R.T.); wucifang@zju.edu.cn (C.W.)
2   Land Academy for National Development, Zhejiang University, Zijingang Campus, Hangzhou 310058, China
*   Correspondence: lilyindutch@zju.edu.cn

**Abstract:** With the aim of improving farmland use efficiency without damaging the social function of farmland, Chinese policymakers have proposed the 'trifurcation of land rights' reform. When it comes to realization of the law, however, neither the Ownership Model nor the Bundle of Sticks Model can adequately explain this reform. The tree concept of property, which provides a new perspective in delineating property rights based on the function served by specific properties, is thus adopted. We find that this tree concept of property helps to better understand and realize the trifurcated rights on farmland in China. Also, a balance between the social and economic functions of farmland can be reached through reconstruction of the property rights involved, a process which comprises three steps: identification of the nature of the newly-established rights; configuration of the rights and duties of relevant entities; and state intervention in the enforcement of relevant rights with the aim of realizing certain social values. Finally, this paper argues that success of this trifurcated structure requires a systematic design of the Chinese Civil Code. In particular, it requests further improvements in legal rules on farmland lease.

**Keywords:** property rights; tree concept of property; bifurcation; trifurcation; farmland lease; reconstruction

---

## 1. Introduction

Rural decline and thus rural revitalization are receiving more attention worldwide [1,2]. From the Common Agricultural Policy (CAP) of the EU to the national strategies of Targeted Poverty Alleviation and Rural Revitalization newly adopted in China, the goal is to provide more opportunities and tools for rural development based on institutional innovations [3,4]. With innovations in (rural) land institutions, such as land policies and relevant legal reform, (other) production factors like capital, labor, and technology may evolve and thrive in rural areas [5]. Although a series of investment and subsidy policies and programs have been adopted to support rural development in China, urban-rural income inequality has been growing rapidly in the past three decades [6,7]. One key factor lies in the weakening of the economic value and function of rural land, caused by restricted transferability of rural collective land rights [8,9]. Legally speaking, the design and arrangement of different rights and duties on land reflect the balance between various interests in land and different functions of land by legislatures. Meanwhile, this balance has to be compatible with the stage of social development as well as the social goals and values pursued [10,11].

Collective land ownership in rural China was originated and affected by communism and the former Soviet Union. Due to the adoption of a 'giving priority to the development of heavy industry' strategy after the foundation of the People's Republic of China (PRC) [12], farmers' private land was pooled together in cooperatives through the agricultural cooperative movement, to make it easier for the

state to acquire food from rural areas to support industrial development [13]. With the reinforcement of China's household registration management system (the HRMS, or *hukou* in Chinese) implemented from the mid-1950s, farmers were directly tied to their land [14]. The full implementation of the Household Responsibility System in rural China since the beginning of the 1980s, and the following individualization of rights to contract and manage (agricultural) land, (the RCML) facilitated the (mostly temporary) moves of rural people, especially with the recognition and permission of (contracted) land transfer in central policies since 1984 [8].

Legalization of the RCML in the 2003 Rural Land Contracting Law (the RLCL) and the 2007 Property Law (the PL) has strengthened farmers' private land rights. However, transfer of this individualized land right remains legally restricted. Reasons behind this restriction lie in the contradiction between the social security function and the economic function imposed on rural collective land [8,9]. On one hand, due to the lack of most social security services in rural China, contracted farmland has long been the most basic and important means of production that provides life support for farmers [14,15]. On the other hand, the increasing number of migrant (rural) workers and increasing areas of idle or even abandoned farmland require mechanisms like market transfer to ensure that the land is being used by those who are still engaged in agricultural production to improve land use efficiency [15]. In this context, central policies have first lifted restrictions imposed on the transferability of contracted farmland by setting up or allowing relevant local experiments [8]. Second, in order to improve land use efficiency on the basis of secured land rights of farmers, the central government proposed the 'trifurcation of farmland rights' reform [15,16] as a top-level design [17]. The reform aims to strike better balance between these two contradictory functions of farmland based on a further split of the RCML into two separate rights. One focuses on securing collective farmers' land use rights attached to their collective membership, and the other focuses on the right of other entities to use land. Despite the questions and doubts regarding the applicability and legality of this trifurcation reform [16,18], legalization of the three separate rights is taken seriously by the central legislature as a political goal. The question is how this new delineation of rights on farmland can be better defined and reflected in the current legal system.

From a historical perspective, property rights on land worldwide initially had strong social character [9,11]. Yet with the modernization of national governance, the economic function of land has grown, while the social character of property rights has decreased. From a legal perspective, the interactions and balance between the social and economic functions of land can be explained at the macro, mezzo and micro levels, centering on a resource-specific analysis of property entitlements desired by the tree concept of property [10]. This property tree framework provides a new perspective for delineating property rights based on the function served by specific properties. Combined with the differences in how property rights are delineated in the common law system and the civil law system, this paper proposes a three-step analysis of the balance between the social and economic functions of farmland in China through restructuring the property rights involved. This analytical framework deepens the understanding of the 'trifurcation of farmland rights' reform on one hand; on the other hand, it necessitates a systematic reconstruction of China's farmland rights system, which can be realized in the final Chinese Civil Code.

## 2. The Theoretical Basis

### 2.1. Interactions between the Social and the Economic Functions of Land

Regarding the function of land, it first serves as a store of wealth and provides production of food, fiber, fuel or other biotic materials for human use [19]. In other words, it functions to safeguard basic human needs for food and shelter. Also, this social function of land has related closely to public interests, such as equal access to land use, since ancient times [20,21]. Economic functions of land as a commodity emerge and evolve with the urbanization and industrialization process, and have a close relationship with its location, land use purpose (determined by zoning/planning), supply (by government), and other

regulatory measures. Among the contradictions between different functions of land, the one between its social and economic functions is most highly disputed. From the perspective of national governance, Deng [11] proposes that—as opposed to the West, where property rights primarily have economic and political functions—-property rights in China also have a strong social character, which was the key to the millennial continuity of Chinese agrarian civilization. With the modernization of national governance, these functions interact with and transform each other. In particular, the state's ability to supply public goods, as one dimension of national governance, is directly proportional to the economic function of land (property rights); and inversely proportional to its social function.

Deng's observations on the changing character of China's land property rights from a historical perspective are inspiring. However, he did not consider that property rights in the West, especially in Europe, also had strong social attributes across history. The most obvious manifestation of this social character is the personal servitude system that existed in the civil codes of several European countries. Personal servitude originates from ancient Roman law. It was designed to last through the life of certain beneficiary, like one's widow; or for a shorter term, such as 'for the widow until she remarries.' Personal servitude mainly comprised three types of rights: *usufruct*, a right to use property and take its yields; *usus*, a right to use property without its yields; and *habitatio*, a right to occupy a dwelling. The personal servitude ran with the beneficiary and was terminated upon his/her death [22]. As an important legal instrument for ensuring the wellbeing of certain individuals, personal servitude satisfied the social need of the impoverished as the (modern) welfare state had not yet emerged until the late 19th century. At that time, the division between land/predial servitude (or easement) and personal servitude was upheld and regarded as a basic principle of property law (civil code) in most European countries as successors of Roman law. However, due to its linkage with feudal privilege and the need for improving the marketability—-that is, the economic function—-of real property, this division became blurred or was even abolished in an increasing number of civil codes, for example, the French Civil Code and the Dutch Civil Code [23].

Although some (European) countries (e.g., Germany, Austria, France, Italy) have kept the usufruct, *usus*, and especially *habitatio* in their laws, they are rarely used in practice, and personal servitude as a title has been mostly discarded. One exception is the restricted personal servitude (Beschränkte persönliche Dienstbarkeit) in the German Civil Code (SS. 1090-1093). Like the old system, restricted personal servitude may only be created for the benefit of a certain (natural or legal) person. When the right-holder dies/bankrupts, the servitude terminates. German Civil Code also contains another special type of limited rights of use called *Reallasten* (encumbrances/charges on land), which entitles certain right-holders to receive regular benefits from land [23]. It originates from the *altenteilsrecht* (in old Germanic customary law), which was often connected with support for elderly and retired farmers through the transfer of farm ownership to their children or successors (the main features of *Reallasten* include: the holder has the right to claim regular payment of certain yields from the land, instead of a right to occupy and use the land; it is based on recurring acts of payment by landowner to the right-holder; and the payment may only be made in yields from the burdened land). As a real right or right *in rem*, *Reallasten* suited the need of an agricultural society as it provided a stable livelihood for retired farmers. After the country's entry into modern society characterized by an (post-) industrial economy and a higher level of urbanization and social securities, *Reallasten* is rarely used in real life. It can be said that the evolution of the personal servitude system in Europe has revealed interactions between the social and economic functions of land from a national governance perspective.

### 2.2. Social Duties of Property Owners Based on the Tree Concept of Property

The discussions above clearly delineate the relationship between land functions and national development at the macro level. The tree concept of property, on the other hand, provides a deeper observation of the balance between the intrinsic nature of property rights as private rights and the extrinsic regulation by the state in the name of public interest at the micro level, or—-in this paper—-at the mezzo level. Different from the Ownership Model in the civil law system and the Bundle of Sticks

Model in the common law system, both of which center on the absolute rights of right-holders, this new tree concept holds that property comprises analytically distinct entitlements that government may reshape for regulatory or redistributive purposes (Table 1). In addition to the individualist element—-the autonomous-control rights (instead of exclusion)—-of property, the social element—-the social function of property—-is another component of the trunk of the property tree, or the essence of property [10]. In other words, this tree concept of property provides a new perspective for defining property rights, which is based on the function served by specific properties.

**Table 1.** Comparison between the tree concept, the ownership model, and the bundle of sticks model of property.

| Types of Models<br>Aspects | Ownership Model | Bundle of Sticks Model | Tree Concept of Property |
|---|---|---|---|
| Time and place of emergence | Since French Code Napoleon at the beginning of 19th century | The early 20th century in the US | 1930s in Italy |
| Definition of property | Exclusive, single, indivisible, and different in nature from lessor property interests | Pluralistic and fragmented, a set of distinct entitlements | Property comprises analytically distinct entitlements that government may reshape for regulating purposes |
| Nature of property | A relation between a person and a physical thing | A relation among persons concerning a thing | The trunk of property tree includes autonomous-control rights and social function |
| Structure of property | Exclusion (too simplistic) | A meaningless discussion | Complex: the core of property is more than exclusion, it is use governance |
| Property and value pluralism | Fundamental value: autonomy | Has little to say about values | Places the question of property's values front and center |
| Owner's duties | Allows for minimal duties | Does not emphasize duties | Property entails (both negative and positive) duties on owners |

First, the inclusion of the social function of property in the trunk of the property tree shows that property entails duties on the part of owners. In addition to negative duties not to harm others, it also includes positive duties to share certain resources [10]. The Constitution of the German Reich, usually known as the Weimar Constitution, was issued in 1919 and is the first constitution in Europe to provide that 'Property entails obligations. Its use shall also serve the public good' (Article 153, Paragraph 3). The current constitution—-Basic Law for the Federal Republic of Germany—-directly inherits this regulation in its Article 14 (Paragraph 2). Constitutional confirmation of the social function of property is even older in Latin America, where it was instrumental in justifying the agrarian reforms aimed at achieving fairer distribution of land in several countries in the region. For example, the 1917 Mexican Constitution (Article 27) empowered the state with the right to impose such limitations as public interest on private property. By the middle of the twentieth century, the Social Function Doctrine had been explicitly incorporated into most Latin American constitutions [21]. Evidence regarding agrarian reform in Brazil and elsewhere has proven that selective redistribution of large unproductive landholdings guided by the social function principle may promote investment and growth by creating additional incentives for using land productively [24]. Property rights of specific resources can be reshaped for regulatory purposes according to this new concept of property.

Second, the tree concept of property has a pluralistic nature, which facilitates debate regarding which values ownership of specific resources should promote. Specifically, by insisting that the core of property is more than exclusion, it is use governance (governing relations among multiple owners and users of resources, rather than excluding non-owners), the tree concept restored owners' negative freedom to the discourse of property. Then, by qualifying an owner's use-control entitlements with the social function of property, it emphasized the need to balance, or fit, negative freedom with the other (public) values of property. Finally, by grounding the social function in the many resource-specific branches of property, the tree concept eased the problem of fitting competing values. In other words, by grounding values and interests in the context of specific resources, the tree concept of property may guide lawmakers to weigh conflicting values involved in different resources [10].

*2.3. Resource-Specific Analysis of Property Entitlements at the Micro Level*

This new resource-specific delineation of property rights set forth by the tree concept of property relies on a detailed analysis of legal relations among the actors involved. Taking the agricultural land —one branch of the property tree—as an example, in countries where agriculture still counts for rural and national development, productive efficiency and more equal access to land are the dual goals set on the land. Landowners or entitled leaseholders have a duty to make efficient use of land; and if they do not, they may be compelled to transfer the right to use land to the landless or land-lost people [10]. Furthermore, whether the property concerned is under private, public, or common ownership, the specific rights and duties involved vary based on the particular location [25].

Overall, analysis of the interactions and balance between the social and economic functions of land at the macro, mezzo and micro levels described above facilitates understanding of the 'trifurcation of land rights' reform in China (Figure 1). At the macro level, with the emergence of the welfare state and modernization of national governance, need for the economic function of land stimulates privatization and individualization of farmland rights. At the mezzo level, a balance between strengthened private land rights (the economic function) and the public interest involved (the social function) in farmland use may be struck based on the tree concept of property. Private property rights and entitlements may be reshaped for regulatory purposes. At the micro level, as the social values attached to individual resources change, a restructuring of property rights occurs [26,27] which requires a resource-specific analysis of property entitlements. The trifurcation of rights on farmland reform in rural China is such a restructuring process that aimed at striking a better balance between the social and economic functions of farmland.

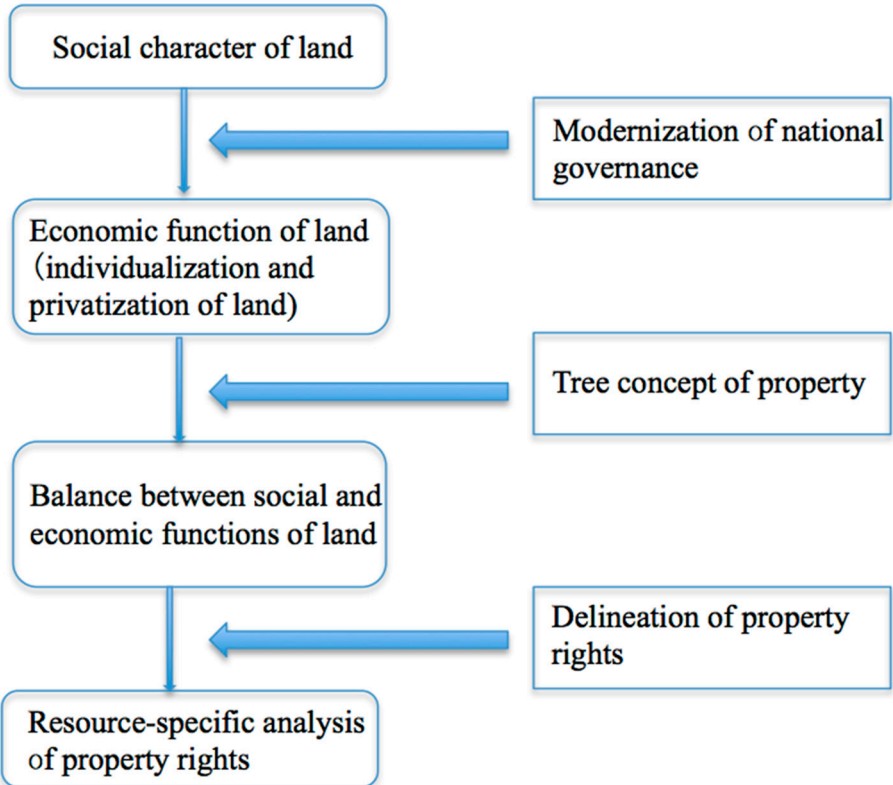

**Figure 1.** A three-level understanding of the "trifurcation of land rights" reform in China.

*2.4. A Three-Step Analysis of the Balance between Functions of Contracted Farmland*

Honoré [28] has first identified nine distinct rights and two duties that regularly apply to the full ownership (or 'fee simple' in the common law system) of land (Figure 2). Lesser property interests (like leasehold) and properties with special features like common-pool resources would be categorized

by the absence of one or more distinct rights and (increased) duties [25]. As argued by Bromley [29], property rights are the result of a social construction process. The social values attached to specific properties affect the arrangement of rights enjoyed and duties afforded by the right-holder [29]. In the meantime, reconstruction of rights and duties facilitates the realization of social values attached to individual properties.

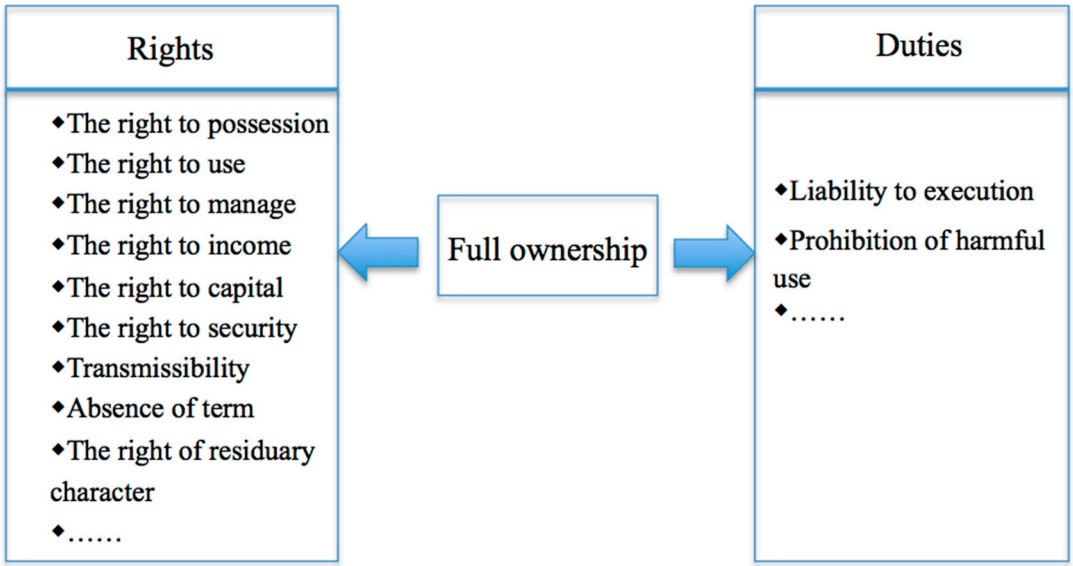

**Figure 2.** Rights and duties that regularly apply to the full ownership of land [28].

It is noteworthy that the analysis and explanation of property rights and entitlements given by Honoré and Bromley primarily apply to countries with a common law system, which does not clearly differentiate (real) property rights from personal rights. Also, as there is no severe limitation from the *Numerus Clausus*, the parties involved may create new (real) property rights outside the law [8]. This arrangement is contrary to the situation in countries with a civil law system, such as China. This research focuses on the reconstruction of rights on rural collective farmland in China, covering both collective land ownership and the right to contract and manage land (RCML) enjoyed by individual households. Generally speaking, the analysis may use Honoré's framework of rights and duties. However, before beginning to analyze specific rights and duties, it is necessary to identify the nature of the right concerned, especially in the case of a non-ownership right. Therefore, when it comes to the question of how the reconstruction of property rights involved may strike a balance between the social and economic functions of contracted farmland in China, a three-step analysis based on the three theoretical bases above is suggested. First, we must identify the nature of the newly established rights. The next step concerns how the rights and duties of relevant entities are configured. Lastly, we must consider state intervention in the enforcement of rights involved, with the aim of realizing certain social values (see Figures 3 and 4 below).

## 3. Changes in the Structure of Property Rights on Contracted Farmland in China

### 3.1. Bifurcation of Rights on Contracted Farmland

Before the foundation of the PRC in 1949, private land ownership was adopted due to the tradition of ancient China (since Qin Dynasty) and the need to attract farmers to participate in the revolution [30]. The Platform of Chinese Land Law (*zhongguo tudifa dagang*), the most significant land law before the Communist Party of China (CPC) came into power, declared that China would accept the 'land to the tillers' agricultural system as a clear principle of Chinese land law (Article 1). However, the rapid collectivization movement that began in the early 1950s soon annihilated private ownership. For contracted farmland, the following process of change shows precisely how private land ownership

was transformed into collective ownership: farmers' ownership and utilization in the time of the land reform movement (1950–1952); farmers' ownership and collective utilization during the junior agricultural cooperation movement (1951–1956); inchoation of collective land ownership during the senior agricultural cooperation movement (1956–1958); the stage of collective ownership and the utilization of land characterized by 'a three-level ownership of collective land on the basis of production teams' (1962–1978); and finally the collective land ownership and farmers' utilization of land under the HRS (1978 until now). It is noteworthy that the meaning of collective land ownership in different stages varies, as the nature of the collective organizations to which it is attached is different [8].

In the late 1970s and early 1980s, the HRS replaced the monolithic collective ownership of rural land in the wake of agricultural de-collectivization, and provided Chinese farmers with an individualized land use right [12]. This right was prohibited from transfer during the first few years [30]. A change was initiated and has been safeguarded by a series of central policies since 1983. In the General Principles of the Civil Law of the PRC promulgated in 1986, the formal term of 'Right to Contract and Manage Land (RCML)' was proposed for the first time by law. Also, since the 'subcontract' (short lease to other collective members) of contracted farmland appeared in the 1984 No. 1 Document of the CCCPC, ability to transfer the RCML was officially confirmed and gradually enlarged by a series of central policies [8]. Nonetheless, it was not until the promulgation of the RLCL in 2002 that the RCML as an independent right of individual households to occupy, use, and seek profits from the contracted farmland and the transfer of it were legalized. Pursuant to Article 32 and Article 42 of the RLCL, the RCML may be transferred through subcontracting, leasing, exchanging, assigning, or being invested in cooperatives or companies. In the Decision of the Central Committee of the CPC on Number of Major Issues on Deepening the Overall Reform issued in November 2013 and the newly-revised RLCL in 2019, mortgage and guarantee of contracted farmland as collateral have also been confirmed.

Following the three-step analysis of the balance between the social and economic functions of farmland through reconstruction of property rights, the RCML separated from collective land ownership is gradually identified as a real property right in the 2002 RLCL and the 2007 PL. In order to improve land productivity, rights (and duties) of individual households—-the right-holders—-have simultaneously been reconstructed. In addition to the absence of term and the right to withdraw contracted land in certain occasions, rights related to the possession, use, profit-making, and transfer of contracted land have been granted to individual households. Duties of both parties have also been readjusted (Figure 3). Theoretically, bifurcation of rights on contracted farmland primarily deals with relationships between the collective landowner and individual households. However, state interventions in the reconstruction of rights on contracted farmland are obvious, and mainly reflected in the restrictions on the transferability of both collective land ownership and individual households' RCML. This is the third level of analysis of the balance between the social and economic functions of farmland under the bifurcated structure. From an economic perspective, this separation of RCML from collective land ownership—-the bifurcation of rights on contracted farmland—-encouraged rural households' incentives for production by giving them freedom of land use rights and decision-making power, and closely connecting rewards with their performance.

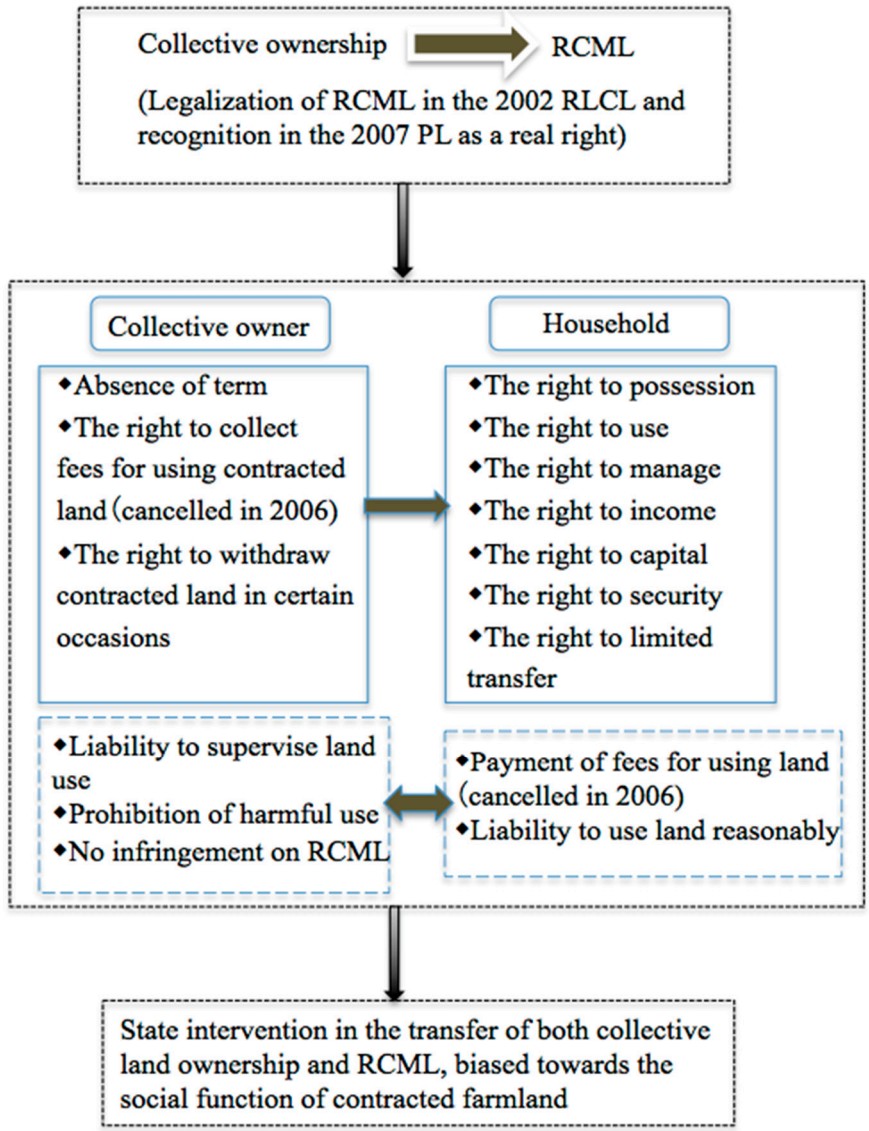

**Figure 3.** A three-step analysis of the bifurcation of rights on contracted farmland.

*3.2. Contradictions between the Social and the Economic Functions of Farmland*

From the beginning of the 1980s, collective farmland and rural homesteads were contracted out and allocated to individual households on an egalitarian basis [31,32]. Also, a periodic reallocation of contracted farmland was adopted to ensure continued equality according to demographic changes in specific households. Despite the pressure to raise agricultural output to fulfill the village grain quota, egalitarianism by and large took priority over efficiency in land rights distribution within villages during the early reform years [32]. The reconfiguration of property rights on contracted farmland between collective and individual households – the bifurcation strategy—-focuses on the fair distribution and social function of farmland among all collective households.

However, the development of the farmland (transfer) market and the national goal of realizing farming at scale require a stabilized right to use farmland. The 2002 RLCL not only confirms the transferability of the RCML, but also imposes a prohibition on reallocation of farmland (Article 27). The increased security of the RCML to some extent promoted the speed and scale of the farmland market [33]. From the early 1980s to the early 1990s, the proportion of farmland in circulation in China has been very small. According to survey data from fixed observation sites in rural China, from 1984 to 1992, the percentage of households that did not transfer farmland has reached 93.8 percent, and that

of households that transferred part of their farmland is only 1.99 percent [34]. Since the promulgation of the RLCL in 2002, the scale and speed of farmland transfer has substantially increased. Rates of farmland transfer in 2003 in the eastern, central, and western China regions were 9%, 11.6%, and 3.86%, respectively; in 2013, however, the numbers increased to 26%, 31%, and 20% correspondingly [35]. Restrictions imposed on the transferability of contracted farmland in the RLCL, with the aim of preventing farmers from being landless [8], have been gradually lifted through a series of central documents since the 2010s. In terms of the form of transfer, however, subcontract and lease of farmland—-characterized by their short-term nature—-were and still are the leading choices for individual households to transfer their contracted land [8,33]. In 2014, proportions of the subcontract and lease of contracted farmland nationwide were 46.53% and 33.17%, respectively. Meanwhile, the area transferred by subcontract and lease accounts for 79.7% of the total transferred area, and those transfers have increased since 2010 by 95.83% and 173.47%, separately [35].

In the meantime, development of industrialization and urbanization has bred a large movement of workers migrating from rural to urban China. The proportion of the rural labor force engaged in agriculture has thus decreased [36], causing problems like non-agriculturalization, non-grain preference, and abandonment of farmland use in rural areas [11,36]. Compared with the national goal of improving scale farming and productivity of agricultural land, farmers prefer to regard contracted farmland as the last resort for their social securities and employment. It seems that there is an inextricable contradiction between more efficient use of contracted farmland and its life-support function for rural collective farmers, and the bifurcation of rights on contracted farmland and the following transfer of RCML cannot resolve this dilemma.

## 3.3. Trifurcation of Rights on Contracted Farmland

In 2014, the central government proposed the reform of 'three rights (ownership, right to contract land, and right to manage land) separation'—-or more accurately, trifurcation of rights on contracted farmland—-to facilitate farmland transfer [15,17]. It is predicted that the productivity of contracted farmland can be improved based on a stabilized and protected right to contract land among collective members and a protected right to manage land among outside and inside investors. The newly revised RLCL, which went into effect on 1 January 2019, finally recognizes this trifurcation of rights on contracted farmland, by confirming the farmland contractors' rights to transfer out their rights to manage land (to other parties by their own decision), in addition to collective land ownership, and their rights to contract and manage land. The right left to contractors is named as the right to contract land (RCL) (Article 9 and Article 36 of the 2019 RLCL).

Before the final confirmation in the 2019 RLCL, there were at least three academic proposals for the arrangement of the trifurcated rights on contracted farmland in China. The first is the 'ownership + collective membership + RCML' structure, where membership originates from collective ownership and covers collective members' right to contract farmland from the collective owners. According to Wu [37], the right to contract land (RCL), like other rights enjoyed by farmers as collective members, is directly generated from ownership, rather than from the right to contract and manage land (RCML). With legal confirmation of the full rights attached to collective membership, including the RCL, the current RCML as a status-based right is thus meant to be transformed into a pure property right. The second is the 'ownership + RCL + RML' structure, where RCL also originates from collective ownership, and RML is the former RCML with no influence from farmers' status [38]. The third is the 'ownership + RCML (with burdens)/RCL + RML' structure, where RML is generated from the existing RCML and can be transferred to third parties as a pure property right. Rights left to the transferor are named as RCL, or RCML with burdens [39,40].

The former two structures may increase the costs of institutional change, because: first, collective membership is not an official right; or put another way, there are no unified and clear rules on collective membership under the current legal system. Second, as the nationwide registration and certification of RCML are almost complete, it is not practical to replace it with a new right and renew the certificate.

Most importantly, the reasoning behind these two structures does not maintain the continuity of the logic behind the bifurcation strategy. Based on the separation of RCML——as a pure property right——from collective ownership and the following legal protection, both individual households' basic living and land productivity can be secured and improved. Then, with the RML splitting from RCML in the case of farmland transfer, a reconfiguration of RCML between collective households and other business entities and a distinction between the social and economic functions of contracted farmland become possible. In essence, the former two structures do not establish a trifurcated structure of rights on contracted farmland, but rather a further explanation of the original meaning of the bifurcated structure. It can be said that the logic, as well as the explanation, of the trifurcated structure provided by the third structure above provides the most accurate fit with the original intention of the central policy.

## 4. The Trifurcated Rights on Contracted Farmland and their Legal Status

### 4.1. A Mixed Influence from the Ownership Model and the Bundle of Sticks Model

It is worth noting that the description of trifurcated rights on contracted farmland in central policies and law, specifically, the split of RCML into a RCL and a RML in the case of farmland transfer, has treated the RCML as a bundle of sticks——the concept of property used in the common law system. This mode of thinking is inconsistent with the guiding ideology of the Property Law of the PRC, whose creation was heavily influenced by German property law. Although the PL as well as the Book of Real Properties of the Chinese Civil Code (Draft) draw few rules directly from the German Civil Code [41], the influences of German law on the ideology, teaching, and drafting of property law in China are significant (see Law of the People's Republic of China, available on: https://en.wikipedia.org/wiki/Law_of_the_People%27s_Republic_of_China, accessed on 7 January 2020). According to the Ownership Model of property in German law——one typical example of the civil law system——the RCML is meant to be a complete and indivisible right. Transfer of contracted farmland, or the RCML to other entities, does not affect the enjoyment of transferors' RCML, which is burdened with certain limits within the period of the transfer contract. When the contract is due, the restricted RCML of the transferor reverts to a complete right. To some extent, rules in the 2019 RLCL on the maintenance of the RCML as an independent right (Article 8) and the separation of RCML into a RCL and a RML in the case of farmland transfer (Article 9) reflect combined influences from the Ownership Model and the Bundle of Sticks Model.

From a global perspective, convergence between these two legal systems is growing, especially when it concerns strategies for verifying titles and limits on the number of property forms [42]. This, however, cannot reduce their fundamental differences in defining property. Comparatively speaking, delineation of property rights in the common law system focuses more on the usage of resources, rather than the dominium of property owners in the civil law system. Adoption of the Bundle of Sticks Model facilitates the trifurcation of rights on contracted farmland, and thus the transfer of land to more efficient users. Nonetheless, with the aim of compiling a unified and coordinated Civil Code of China, it is better to maintain the mode of thinking used in the civil law system——specifically, the German law——and improve land use efficiency through further delineation of the property rights involved.

### 4.2. Relations among the Trifurcated Rights

Under the bifurcated structure, as an independent property right——a usufruct——holders of the RCML are entitled to occupy, use, and make profits from the contracted farmland (Article 125 of the Property Law). Meanwhile, with the aim of promoting scale farming and improving land use efficiency, holders of the RCML are also entitled to dispose of the land through market transfers. A RML may be established for the new land users.

Combined with the new revisions in the 2019 RLCL, relations among the separated rights can be explained through the four routes shown in Figure 4: (1) Route One: with the farmland received from

the collective by contract, if the individual household decides to cultivate on its own, then it enjoys a full RCML. (2) Route Two: if the household intends to transfer its RCML permanently (including assigning and interchange) to other collective members, then the RCML can be transferred to and enjoyed by new contractors as a full and independent right. (3) Route Three: if the household decides to transfer its contracted farmland to other business entities within a certain period (including lease, being invested in cooperatives/companies, and mortgage), a RML will be separated from the RCML and transferred to the business entities concerned. According to Article 9 of the 2019 RLCL, after the RML is transferred to other entities, the transferor still enjoys a right to contract land (RCL). In essence, however, this RCL can be regarded as the original RCML with burdens, meaning the transferor cannot enjoy and exercise rights related to the transferred RML within the period of transfer contract. Hence, there is no need to certify and register a separate RCL for the transferor in the case of farmland transfer. (4) Route Four: when the RML of the transferee terminates because the transfer contract becomes due or for other reasons, the transferors' RCML with burdens will revert to a full right.

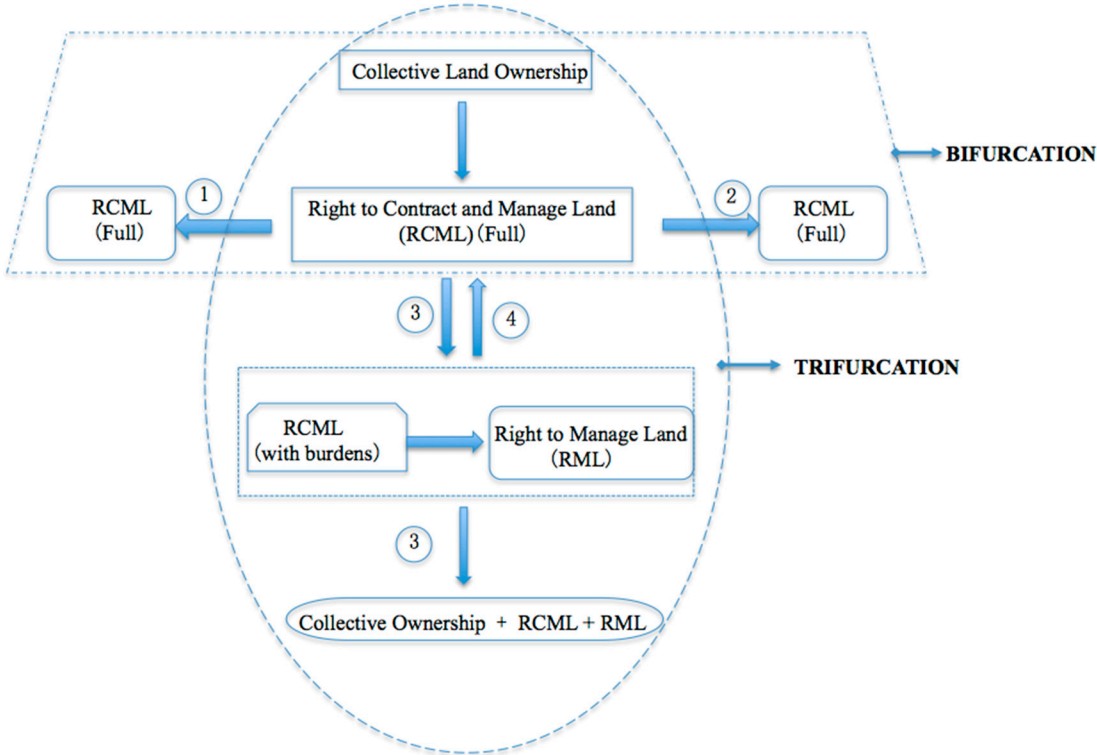

**Figure 4.** The trifurcation structure of rights on contracted farmland and relations among them.

Based on the analysis above, we believe that the 'ownership + RCML (with burdens) + RML' structure should be a more convincing explanation for the trifurcated rights on contracted farmland in rural China. In addition to the cost-saving change of institutions, it secures and respects individual households' choices not to transfer out their contracted farmland (permanently or for a long-term), by keeping the existing RCML as a full and independent right. Furthermore, it strikes a better balance between the social and economic functions of contracted farmland.

### 4.3. Nature of the RML—-the Third Right in the Trifurcated Structure

The main purpose of trifurcated rights on contracted farmland is to promote land use efficiency without damaging collective households' rights to land. As argued above, in the trifurcated structure, the RCML is supposed to be a pure property right, which may secure both the social and the economic function of contracted farmland for the collective households. Realization of the social function relies on a fair and clear definition of collective membership, which covers the right of collective members

to contract farmland (RCL) from the collective. With an independent RCML, the landholders may make full use of the land on their own, or transfer it out to other entities. The RML as the new right to use transferred land aims to improve land use efficiency, that is, the economic function of land. Theoretically, it is a pure and independent property right like the RCML. According to the 2019 RLCL, the RCML can be directly assigned, exchanged, leased, mortgaged, and invested to cooperatives or companies by individual households without being segmented. Moreover, in the cases of lease, investment in cooperatives or companies, and getting a mortgage, a RML is established for transferees to differentiate it from the RCML of the transferors.

Specific to the nature of the RML, in accordance with the original purpose of the trifurcated structure for rights on contracted farmland, with the transferred RML, transferees—-the other business entities—-should be able to make long-term and sound investments in farmland. A real property right, in this case, may better realize this goal because an exclusion right lies at its core [10]. This, however, contradicts the nature of the leasing right as a personal right in the case of farmland lease. Meanwhile, when individual households choose to become cooperative members or company shareholders by investing in the RML, it is suggested that the household transfer its RML through lease first, and become members or shareholders after the cooperative or the company makes stable profits. It can be said that, aside from mortgage, RML is mainly established through lease. In practice, the main form of farmland transfer is lease, including subcontracts (short leases to other collective members), most of which have terms of less than 5 years [43]. Also, lease is used more often between farmers and other collective members (non-families), farmers and their relatives, farmers and other private persons, etc. [44,45]. The RML is, first and foremost, a personal property right. In the meantime, promotion and protection of long-term lease of (contiguous) farmland is necessary to attract external investment in agriculture and the countryside, especially with the implementation of the latest Rural Revitalization Strategy (see Opinions of the CPC Central Committee and the State Council on Implementing the Rural Revitalization Strategy (the No. 1 Document) issued on 2 January 2018). Based on article 41 of the 2019 RLCL, in the case that the RML is transferred for more than 5 years, both parties of the transfer contract may require registration, otherwise it may not challenge any bona fide third party. Like the RCML, the RML becomes valid when the (transfer) contract becomes effective. It is not compulsory for the parties to register the transferred RML in local governments. Instead of establishing a constitutive or compulsory registration system for the transfer of urban land, a consensual or a voluntary registration system is adopted for the transfer of RCML [46], or for the (long-term) RML. To some extent, a long-term RML can be delineated as an *in rem* right of a limited scope under certain conditions.

However, if we take a closer look at the other relevant rules on RML in the 2019 RLCL, it is hard to come to the same conclusion. First, unlike the RCML, rules on the liabilities for infringing upon the RML are limited to liability for breach of contract (Article 56 and 59). On the contrary, Article 57 of the RLCL provides that the contract-issuing party infringing upon the RCML of individual households shall bear civil liabilities like cessation of infringements, return of original objects, rehabilitation of original condition, removal of obstacles, elimination of dangers, or compensation for losses. These are the liabilities for damage to real property rights. Second, in the case that individual households become cooperative members by investing their RML, the transferred RML is unstable as members have rights to voluntarily join and freely withdraw from the cooperative (Article 4 of the Law on Farmers' Professional Cooperatives). Most importantly, in accordance with the *Numerus Clausus* adopted by the PL (Article 5) and the General Provisions of the Civil Law of the PRC issued in March 2017 (the first part of the future Chinese Civil Code) (Article 116), the RML is not confirmed as a specific real property right, or more precisely, a usufruct. Therefore, by nature, the RML in the existing law is a personal right that can be given *in rem* protection with registration in the case of a more-than-5-year transfer.

## 5. A Three-Step Analysis Framework for the Trifurcated Structure

Definition of the nature of RML as a personal right that can be given *in rem* protection is the first step of the three-step analysis of the balance between functions of farmland through the 'trifurcation

of land rights' reform. The second step concerns the arrangement of rights and duties of the three entities involved—-the collective landowner, the individual household, and other business entities. In essence, the creation of RML extends the right of individual households with RCML to dispose of the farmland through transfers. From the perspective of the economic function of land, the RML focuses on the distribution of rights and duties between individual households with RCML and other business entities—-new users of farmland. In addition to the right to income (fees for using land paid by new land users) and the right to limited transfer of contracted farmland, rights related to the possession (except in the case of getting a mortgage), use, profit-making, and retransfer of the contracted land have been granted to other business entities (Figure 5). From the perspective of the social function of land, however, collective land ownership also matters in the trifurcated structure, as it embraces all entitlements related to collective membership, including the right to contract farmland (RCL) from the collective which is the precondition for individual households to obtain and realize the RCML.

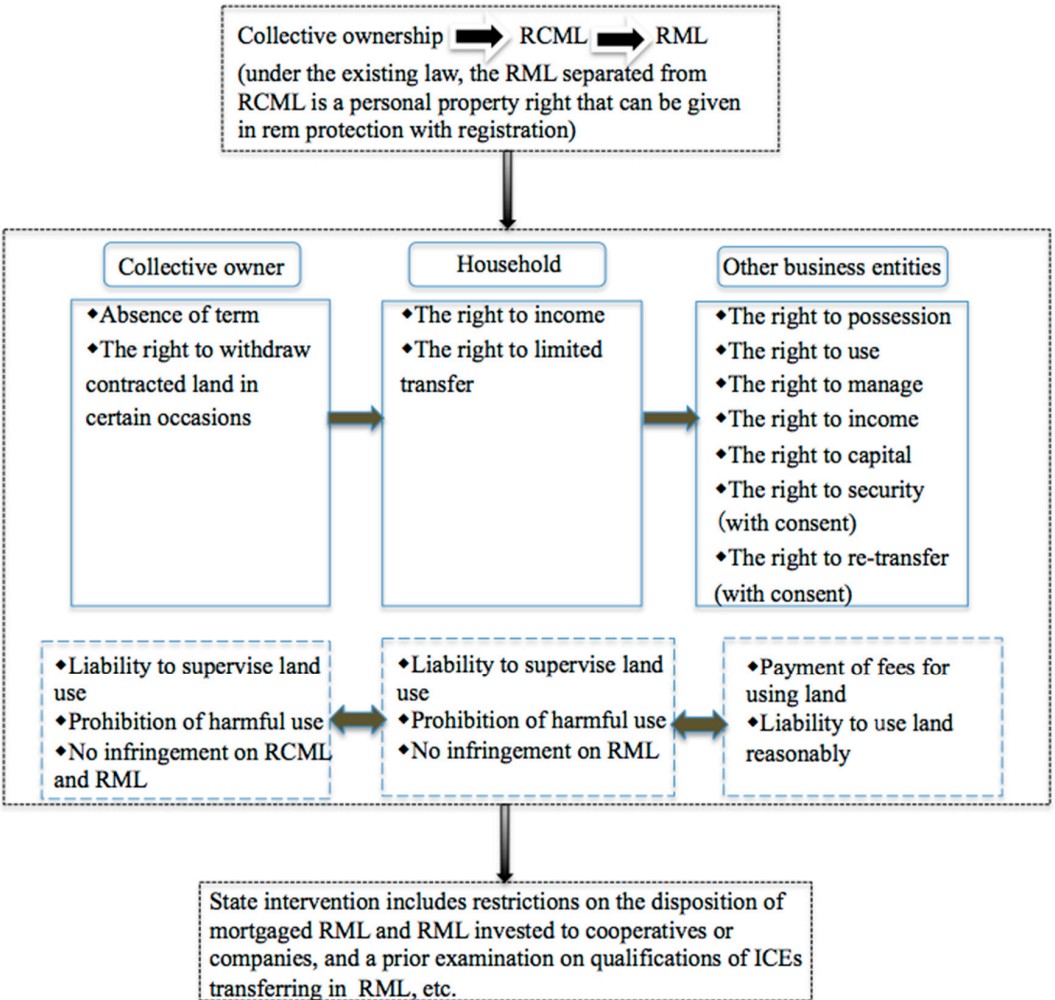

**Figure 5.** A three-step analysis of the trifurcation of rights on contracted farmland.

Figure 5 shows the arrangement and distribution of rights and duties of the three entities in the trifurcated structure. Although the establishment of RML extends the right of individual households to make more efficient use of their contracted farmland, state intervention in restructuring property rights is still obvious. For instance, when an individual household becomes cooperative members or company shareholders by investing their RML into cooperatives or companies, in the case of bankruptcy dissolution of the cooperative or company, the member or shareholder household may repurchase the invested RML in accordance with relevant laws and regulations or the articles of association

of the cooperative and the company (see the Guiding Opinions of the Ministry of Agriculture and Rural Affairs, the National Development and Reform Commission, the Ministry of Finance and Other Departments on Launching the Pilot Program of Investment of Rights to Manage Land in Cooperatives and Companies and Developing Agricultural Industrialization issued on 19 December 2019). If a RML is created through getting a mortgage on the land, in the case that the mortgagor goes bankrupt and cannot return the loan to the mortgagee, the creditor is only entitled to hold the invested or mortgaged RML for a certain number of years. After that, the RML will revert to the household or the collective [8]. As a business with low profits and high risks, government support is indispensable for establishing a mortgage system for the RML, especially support from state-owned financial agencies.

Further, in order to avoid failure of investment in farmland, in the case when the RML has been transferred to business entities like industrial and commercial enterprises (ICEs), local governments at or above the county level should conduct a prior examination of its qualification and the project involved (Article 45 of the RLCL). This requirement aims to protect the RCML of individual households from being infringed by the RML being transferred to (big) business entities. This measure is biased toward the social function of contracted farmland.

## 6. The Bigger Picture—-Restructuring Rights on Contracted Farmland in the Unified Civil Code

### 6.1. Confirmation of the Farmland Lease Contract in the Chinese Civil Code

Under the existing law, RML is primarily created by farmland lease. If the transferee desires a long-term use and investment in land, then a registered lease with *in rem* protection applies. If it is a short-term (less than 5 years) use, or parties of the contract choose not to register a long-term transfer, then the right to use (a personal right) of the transferees can be realized through a leasing contract. Article 40 of the 2019 RLCL provides basic articles for the transfer contract, including the name and residence of both parties; information on the transferred land; duration of the transfer; usage of transferred land; rights and duties of both parties; transfer price; distribution of the compensation for land in the case of expropriation, requisition, and occupation; and liabilities for breaches of contract. Moreover, if the contractor chooses to hire others to cultivate the contracted land for no more than one year, then an oral contract is enough. By and large, in terms of the duration of contract, there are generally three types of farmland transfer contracts: (oral) contracts with a term of less than one year; contracts with a term between one and five years; and contracts with a term of more than five years.

With the aim of improving farmland use efficiency, the availability of various types of transfer contracts, which may secure the freedom of contract of both parties, is one primary issue. Second, equal status for both parties in the transfer contract is meant to be recognized through the rights retained and the duties afforded by both parties. In the 2019 RLCL, on one hand, the transferor has a statutory duty not to unilaterally cancel the transfer contract, except if the transferee seriously breaches the contract—-for example, by changing the agricultural use of land without permission, abandoning land for more than two years, causing serious damage to land, or seriously damaging the land's ecological environment (Article 42). On the other hand, with the consent of the contractor, the transferee may invest in improving soil quality, construct agricultural production facilities, and obtain reasonable compensation for its investment in accordance with the contract (Article 43). More importantly, with written consent of the contractor and records filed with the collective, the transferee may retransfer the RML, including getting a mortgage on the transferred land (Article 46 and 47). These measures may directly facilitate the use of farmland by transferees with the transferred RML. Last but not least, the noncompulsory requirement for registration of the transferred RML reduces the costs of transfer, which also contributes to an efficient use of transferred farmland.

From a comparative perspective, besides land sales market, a rental market based on farmland lease is an indispensable part of farmland transactions in most countries [47]. As a specific contract, rules on farmland lease are mainly from contract law, which is either a separate part of the civil code as in the Netherlands (Book 7 of the Dutch Civil Code), or included in the law of obligations as in the

German Civil Code (Book 2). In China, as there is no sales market available for farmland according to the current law, farmland lease—-including both long-term and short-term lease – is thus important for future agricultural development. Rules on the leasing contract in the current Contract Law (Chapter 13) mainly apply to houses, rather than land. Therefore, later in the unified Civil Code, a special contract for farmland lease is meant to be recognized in the Book of Contract. These new rules on transfer contract in the RLCL above serve as the main content of this section.

*6.2. Strengthening the In Rem Protection for RML in the Civil Code*

As a personal right that can be given *in rem* protection through registration, legal liabilities for infringements on the long-term RML in the 2019 RLCL and the final draft of the Book of Real Properties of the Chinese Civil Code (made public on December 16, 2019) are insufficient. Chapter 3 (from Article 32 to Article 38) of the PL provides rules on liabilities for different types of infringements on real property rights, which should also apply to the long-term RML with registration. Moreover, as the 2019 RLCL has confirmed the mortgage of RML in addition to the mortgage of RCML, it is necessary to further confirm it in the Book of Real Properties of the Chinese Civil Code. Specifically, Article 186 of this draft should clearly include RML in the description of properties that can be used for mortgage. It is noteworthy that rules on the RCML and the RML in the final draft of the Book of Real Properties of the Chinese Civil Code are simpler than those described in the RLCL. It is predictable that the RLCL will remain in effect before it can be totally absorbed into the final Chinese Civil Code.

*6.3. Realization of the Collective Land Ownership in the Trifurcated Structure*

Although the trifurcated structure of rights on contracted farmland focuses on the RML—-a relatively secured right for the transferee to fully use the land—-the collective land ownership cannot and should not be overlooked. On one hand, the collective may supervise the use of land by the new users, in addition to supervising the use of land by the collective household—-the right-holders of the RCML. As the landowner, the collective may curtail illegal use and damage to contracted farmland caused by the collective household, as well as those caused by the new users, if the household fails to fulfill its obligations. Apart from active supervision of land use, changes in the holding of contracted farmland—-including the retransfer of land by the new users—-has to be reported to the collective (Article 36 and 46 of the RLCL).

On the other hand, the collective is entitled to share profits from the transferred land if it is used for a more profitable purpose. With the aim of realizing rural revitalization in China, attraction of outside investments through a smooth (rural) land transfer system is crucial to the whole process. As the manifestation of the economic value of rural land and development of a rural land market, the collective as the landowner in law is meant to share the added value in transferred land.

In the General Provisions of the Civil Law, a rural collective economic organization—-the legal form of rural collectives—-is confirmed as a special legal person (Article 99). Where no village collective economic organization is formed, a villagers' committee may instead perform the functions of a village collective economic organization in accordance with the law (second paragraph of Article 101). With this legal status, the rural collective economic organization or the villagers' committee is entitled to enjoy and exercise collective land ownership, including the management of contracted farmland. In the newly revised RLCL, in the case that social capitals, such as industrial and commercial enterprises, obtain the RML through transfer, the village collective economic organization may charge an appropriate amount in management fees. Recognition of the rural collective as a legal person in law facilitates land transfer and helps avoid the risks involved in the transfer process.

## 7. Conclusions

The 'trifurcation of land rights' reform in China aims to strike a better balance between the social and economic functions of contracted farmland, through further delineation of the bifurcated rights on contracted land. This balance between land functions through restructuring the property rights

involved originates in the tree concept of property. Different from the Ownership Model of property in the civil law system and the Bundle of Sticks Model in the common law system, this tree concept of property focuses on the need to balance owners' autonomy (or negative freedom) with the other values (the social function) afforded by an individual property. It provides a new perspective for defining property rights based on the function served by specific properties. Furthermore, a three-step analysis framework can be established for balancing the functions of contracted farmland in China through restructuring the property rights involved.

As China's legal system is primarily influenced by German law, differentiation between real (property) rights and personal (property) rights is upheld in laws on property. In the bifurcated structure, although individual households only enjoy rights to use land under collective ownership, establishment of the RCML as a real right secures a long-term and stabilized right for them to occupy and use contracted farmland. The trifurcated structure initiated from the 'trifurcation of land rights reform' should follow the logic used in the bifurcation strategy, meaning the third right—-the RML—-is also an independent property right like the RCML. Nonetheless, it can only be a personal property right, which can be given *in rem* protection with registration under the current legal system.

In the trifurcated structure, the RCML of individual households is restricted because part of its entitlements is transferred to new land users through the RML. Other than mortgage, RML is mainly established through lease. Based on the establishment of rights enjoyed and duties afforded by both parties to the leasing contract, the interests of both parties can be maximized. In the case that the lessee chooses to register the (long-term) lease with the public agency, a lease with *in rem* protection is created. In other words, even with a personal right to the transferred land, new users—-or the lessees of land—-are also able to make long-term investments and reap the benefits from land. In addition to the rights enjoyed, as the legal and actual occupiers and users of land, holders of the RML in the trifurcated structure and the RCML in the bifurcated structure have duties to fully and reasonably use land, rather than leaving the property derelict or damaged. State intervention in restructuring the property rights involved is direct, with the aim of striking a balance between the social and economic functions of farmland, or between fairness and efficiency in farmland use. As the dominant means of establishing RML, a farmland-leasing contract should be recognized as a special contract in the final Civil Code. Also, rules on the protection of RML need further perfection in both the RLCL and the Civil Code.

The collective landowner in both the bifurcated and the trifurcated structure, as a legal person confirmed by the General Provisions of the Civil Law, has rights and duties to supervise land use. The collective landowner may also share certain profits from the transferred use of contracted farmland. It is noteworthy that as an independent legal person, the collective may under a proper procedure decide that farmland inside the collective is jointly used and managed by the collective—-the collective households together—-without distribution to individual households. In this case, transferor of the right to use collective farmland would be the collective, rather than individual household.

**Author Contributions:** Conceptualization, L.L. and R.T.; methodology, L.L.; formal analysis, L.L.; resources, R.T.; writing—original draft preparation, L.L.; writing—review and editing, R.T. and C.W.; supervision, C.W. All authors have read and agreed to the published version of the manuscript.

**Funding:** This study is supported by the National Social Science Foundation of China (Grant No. 16ZDA020).

**Conflicts of Interest:** The authors declare no conflict of interest.

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
