# Peer review of "Reconstruction of China’s Farmland Rights System Based on the ‘Trifurcation of Land Rights’ Reform"

_land, doi:10.3390/land9020051_

Round 1

Reviewer 1 Report

Thank you for letting me review this paper which concerns important issues of Reconstruction of china's farmland rights system based on the "trifurcation of land rights" reform. I have some comments and suggestions for revision which are presented below.

Point 1: I recommend to the authors in the introduction part to add more literature review in order to emphasize the need and originality of your research.

Point 2: From lines 37-40, Legally.....values pursued. () needs source.

Point 3: From line 49-50, this paper proposed.....involved. write this sentence at the end of the introductory section showing the purpose of the study.

point 4: From lines 62-68 and line 72-80 length of a sentence be too much. Make it short.

Point 5: I recommend that you cite more literature review to emphasize the depth of the study. For example Lines 36-37, lines 42-46, lines 51-53 only cite one literature.

Point 6: I suggest that use the same style for citation. Your citation style is mixing on some of them write only in number by excluding page number but in the others, you write both number and including page number. anyways the format of MDPI is used only the number for citation. Please correct and revise all of the citations.

Point 7: From page 83-92, no need for writing such sentences, because only explain like a table of content. Delete these sentences and instead write the purpose of the research in an elaborated way.

Point 8: From lines 111-119 you used long sentences and make it short.

Point 9: Lines 119-127,...Dutch Civil Code (). needs source.

point 10: I recommend that all figures have inserted as a picture. Please change and do in Microsoft office in the menu section insert and the shapes. do like that.

Point 11: Write the discussion section and conclusion section in different titles. Your writing is mixing the two. In the discussion section writes your findings related with other similar literature and conclusion section by revising most fundamental things from introduction, method, and results and then write some recommendations.

Having said these, I hope the authors are willing to revise the manuscript, and I would be willing to revise it again.

Author Response

Point 1: I recommend to the authors in the introduction part to add more literature review in order to emphasize the need and originality of your research.

Response 1: Thank you for the comments and suggestions. The authors have added more literature in the introduction part to emphasize the need and originality of this research.

Point 2: From lines 37-40, Legally.....values pursued. () needs source.

Response 2: The source here has been added.

Point 3: From line 49-50, this paper proposed.....involved. write this sentence at the end of the introductory section showing the purpose of the study.

Response 3: Thank you for the suggestion. Structure of the introduction part has been changed.

point 4: From lines 62-68 and line 72-80 length of a sentence be too much. Make it short.

Response 4: Thank you for the suggestion. The sentences that you mentioned have been shortened.

Point 5: I recommend that you cite more literature review to emphasize the depth of the study. For example Lines 36-37, lines 42-46, lines 51-53 only cite one literature.

Response 5: Thank you for the suggestion. More literature has been added in places you mentioned above.

Point 6: I suggest that use the same style for citation. Your citation style is mixing on some of them write only in number by excluding page number but in the others, you write both number and including page number. anyways the format of MDPI is used only the number for citation. Please correct and revise all of the citations.

Response 6: All of the citations have been revised.

Point 7: From page 83-92, no need for writing such sentences, because only explain like a table of content. Delete these sentences and instead write the purpose of the research in an elaborated way.

Response 7: This part has been deleted, and replaced with a new paragraph.

Point 8: From lines 111-119 you used long sentences and make it short.

Response 8: Thank you for the suggestion. The sentences that you mentioned have been shortened.

Point 9: Lines 119-127,...Dutch Civil Code (). needs source.

Response 9: The source here has been added.

point 10: I recommend that all figures have inserted as a picture. Please change and do in Microsoft office in the menu section insert and the shapes. do like that.

Response 10: All figures have been inserted as a picture.

Point 11: Write the discussion section and conclusion section in different titles. Your writing is mixing the two. In the discussion section writes your findings related with other similar literature and conclusion section by revising most fundamental things from introduction, method, and results and then write some recommendations.

Response 11: Thank you for the suggestion. According to your definition of these two sections, part 6 of the current manuscript is actually the discussion section. The last part then serves as the conclusion section. Therefore, the authors have simplified the title of the last section. 

Having said these, I hope the authors are willing to revise the manuscript, and I would be willing to revise it again.

Reviewer 2 Report

1.  This is an extremely interesting paper. However, it is complex and dense, and needs maximising simplification wherever possible.

2.  On language, good, but revisit these lines for meaning -

104 - do you mean 'is the most disputed'?

178-9 - unclear

189 - do you mean cultivated land? Agricultural land includes pastures.

202 ' may be stricken'? do you mean 'may be striked out'?

249 - 'inchoation'? do you mean something 'begins' or ' a process of becoming more opaque'?

302 - change 'scale farming' to 'farming at scale'

452 - 'may not challenge' - do you mean 'may not be challenged by a third party

etc. etc.

Also wonder if it might be easier for many readers to understand if your use the term 'rights and responsibilities'?

3. Explain how Chinese law adopted mainly German civil law constructs. Via Ottoman Empire  influences into Asia or or earlier?

4. Unclear why there are no clear 'rules' for how a village or other collective managed and governs land; would have been helpful to have concrete examples of where the code needs simple procedures to express this. Seems like to whom a collective member/household may sell its land is a critical governance indicator, especially to give priority to relatives, then other members of the village, before to outsiders etc?

5. Related, I would have like to have seen a few speculations on where the new system will take rural farmers - some figures of rural landlessness would be interesting. 

6. As an observation only I find it interesting that 'collective' tenure is still not defined as a property right but as virtual non-property. This trajectory is different from that evolving in new African and some Asian laws where the owners - i.e. the community members in common - decide the nature of the property, if and when it can be transferred or entirely sold even, but it always acknowledged as its property, and cannot be interfered with. Your vision, as indeed the China vision seems to be, is deeply western in the notion that property only exists if it is saleable and indeed potentially entirely alienable, rendering modern collective ownership weak. Whereas the trend in Africa and Latin America is that there are different types of property, with different incidents or attributes, but share a common nature as under the control of a defined owner/holder. 

Author Response

This is an extremely interesting paper. However, it is complex and dense, and needs maximising simplification wherever possible.

Response 1: Thank you very much for your compliment and suggestions.

On language, good, but revisit these lines for meaning -

Response 2:

104 - do you mean 'is the most disputed'?

Response: Yes, thank you for the correction.

178-9 - unclear

Response: Meaning of this sentence is explained by the following content of this paragraph, which is summarized in three points (specifically, …; then, …; finally,…).

189 - do you mean cultivated land? Agricultural land includes pastures.

Response: Here we use agricultural land as an example to explain the necessity of a detailed analysis of legal relations among actors involved in specific property. It does not specifically refer to cultivated land.

202 ' may be stricken'? do you mean 'may be striked out'?

Response: ‘To strike a balance between A and B’ is a right collocation in English, so here we use ‘stricken’.

249 - 'inchoation'? do you mean something 'begins' or ' a process of becoming more opaque'?

Response: The former understanding is accurate.

302 - change 'scale farming' to 'farming at scale'

Response: Changed. Thank you for the correction.

452 - 'may not challenge' - do you mean 'may not be challenged by a third party

Response: The use of 'may not challenge' is right here.

Also wonder if it might be easier for many readers to understand if your use the term 'rights and responsibilities'?

Response:As legal terms, the use of ‘rights and duties’ is used more often than ‘rights and responsibilities’.

Explain how Chinese law adopted mainly German civil law constructs. Via Ottoman Empire influences into Asia or earlier?

Response 3: China’s legal system is largely a civil law system, reflecting the influence of Continental European legal systems, especially the German civil law system in the 19th and early 20th centuries (source:Law of the People's Republic of China - Wikipedia).

Unclear why there are no clear 'rules' for how a village or other collective managed and governs land; would have been helpful to have concrete examples of where the code needs simple procedures to express this. Seems like to whom a collective member/household may sell its land is a critical governance indicator, especially to give priority to relatives, then other members of the village, before to outsiders etc?

Response 4: In rural China, as collective farmland is contracted out to individual households, emphasis of the current legal system is put on private land rights of individual households. Collective land ownership is thus overlooked to some extent. In this research, we argue that as one branch of the trifurcated structure of rights on contracted farmland, collective land ownership needs to be strengthened. On the one hand, the collective may supervise the use of land by new users, in addition to the supervision for the use of land by collective households. On the other hand, the collective is entitled to share profits from the transferred land if it is used for a more profitable purpose. In the General Provisions of the Civil Law of the PRC, which will be the first part of the final Civil Code, rural collective economic organization and villagers’ committee are recognized as legal forms of rural collectives. More rules concerning the rights of collective landowner are needed in the final Civil Code.

Related, I would have like to have seen a few speculations on where the new system will take rural farmers - some figures of rural landlessness would be interesting.

Response 5: Purpose of the ‘trifurcation of land rights’ in China is to improve the farmland use efficiency through encouraging farmers who are not able or willing to do farming to transfer their contracted farmland out to other farmers or business entities. Also, this transfer cannot damage the land rights and interests of collective households or farmers. On the basis of the three separated rights – collective land ownership, RCML, and RML, rights and duties of the three parties involved are clarified. In practice, the main form of farmland transfer is lease, including subcontract (short lease to other collective members), most of which are less than 5 years. Therefore, the number of rural landlessness caused by land transfer is not big.

As an observation only I find it interesting that 'collective' tenure is still not defined as a property right but as virtual non-property. This trajectory is different from that evolving in new African and some Asian laws where the owners - i.e. the community members in common - decide the nature of the property, if and when it can be transferred or entirely sold even, but it always acknowledged as its property, and cannot be interfered with. Your vision, as indeed the China vision seems to be, is deeply western in the notion that property only exists if it is saleable and indeed potentially entirely alienable, rendering modern collective ownership weak. Whereas the trend in Africa and Latin America is that there are different types of property, with different incidents or attributes, but share a common nature as under the control of a defined owner/holder.

Response 6: Thank you for the sharing of your thought above. In China, the situation is also changing now. With the progress in property rights reform of rural collectives, all of the capitals, assets, and resources owned by specific collectives are identified, appraised, and registered. Combined with the change of collective owners into joint stock cooperatives, rights and interests of collective members to collective properties are secured. This reform has been conducted in 15 provinces, and it is expected to be implemented across China in 2021.

Round 2

Reviewer 1 Report

Dear Authors,

Thank you for the revisions you made to the manuscript! you met my comments really well, and I think it is now ready to be published.